# The Baseline Nutritional Status Predicts Long-Term Mortality in Patients Undergoing Endovascular Therapy

**DOI:** 10.3390/nu11081745

**Published:** 2019-07-29

**Authors:** Keiko Mizobuchi, Kentaro Jujo, Yuichiro Minami, Issei Ishida, Masashi Nakao, Nobuhisa Hagiwara

**Affiliations:** Department of Cardiology, Tokyo Women’s Medical University, Tokyo 162-8666, Japan

**Keywords:** peripheral artery disease, nutrition, endovascular therapy, all-cause mortality

## Abstract

Introduction: Peripheral artery disease (PAD) occurs at an advanced stage of atherosclerosis and its comorbidities are associated with poor prognoses. Malnutrition is related to the severity of atherosclerosis in patients with cardiovascular disease and it predicts mortality. The Controlling Nutritional Status (CONUT) score is calculated from serum albumin concentration, peripheral lymphocyte count and total cholesterol concentration, and it robustly represents the nutritional status of hospitalized patients. This study aimed to determine the prognostic value of the CONUT score in patients with peripheral artery disease (PAD) who were undergoing endovascular therapy (EVT). METHODS and RESULTS: This study included 628 PAD patients who underwent EVT between 2013 and 2017 and were assigned to low (CONUT score 0: *n* = 81), mild (CONUT score 1–2: *n* = 250), moderate (CONUT score 3–4: *n* = 169), and high (CONUT score ≥ 5: *n* = 128) risk groups. The study’s primary endpoint was any death. Patients in the groups with higher CONUT scores were more likely to have chronic kidney disease (*p* < 0.001), impaired left ventricular ejection fractions (*p* < 0.001), and critical limb ischemia (*p* < 0.001) on admission. During follow-up, 95 patients (15%) died. Kaplan–Meier analyses revealed that the patients with higher CONUT scores had lower survival rates (*p* < 0.001; log-rank trend test). Multivariate Cox regression analyses showed that following adjustments for the confounding factors, a higher CONUT score was significantly associated with any death (hazard ratio, 1.15; 95% confidence interval, 1.03–1.30). CONCLUSION: The simple index CONUT score at the time of EVT may predict long-term mortality in PAD patients.

## 1. Introduction

Peripheral artery disease (PAD) occurs at an advanced stage of atherosclerosis, and its risk factors are associated with poor prognoses. The presence of risk factors of atherosclerosis including diabetes, hypertension, dyslipidemia, and smoking habit is developing the status of PAD, coronary artery disease (CAD), and ischemic stroke [1,2,3,4]. However, over 50% of PAD patients are asymptomatic [4,5]; therefore, they do not receive preventive intervention. Successful endovascular therapy (EVT) can restore blood flow to the distal regions of the limbs, but it cannot overcome the high adverse event rate, and the prognostic indicators for patients with PAD after EVT remain unclear. Recently, malnutrition on admission has been shown to be an independent predictor of mortality, and it is related to the severity of the atherosclerosis in patients with coronary artery disease (CAD) and in those with heart failure [6,7,8,9]. The evaluation methods of nutritional status in patients suffering various diseases have been proposed as the Controlling Nutritional Status (CONUT) score and Prognostic Nutritional Index (PNI) [10,11,12,13,14]. The CONUT score is calculated from the serum albumin level, peripheral lymphocyte count, and the total cholesterol (TC) concentration. This score can be calculated only by the blood test, and it robustly represents the nutritional status of patients with CAD [10,11,12,15]. Accordingly, the CONUT scoring system is superior in terms of its convenience and had comparable prognostic accuracy compared to the PNI scoring system [16]. Based on the background above, we hypothesized that malnutrition is associated with poor clinical prognoses in patients with PAD who progressively develop arteriosclerosis. The aim of this study was to determine the CONUT score’s prognostic impact on patients with PAD who were undergoing EVT.

## 2. Materials and Methods

### 2.1. Study Design and Study Population

This single-center cohort study included consecutive patients who underwent EVT between April 2013 and June 2017. Patients were excluded if the laboratory data required to calculate the CONUT score were missing. The CONUT score was calculated based on a patient’s serum albumin level, peripheral lymphocyte count, and TC level (Appendix A). On admission, the enrolled patients were assigned to 4 risk groups based on their CONUT scores, as follows: low-risk group (CONUT score = 0), mild-risk group (CONUT score = 1–2), moderate-risk group (CONUT score = 3–4), and high-risk group (CONUT score > 5) [8]. A baseline CONUT score of >2 is defined as malnutrition, and among hospitalized patients who had acute heart failure or CAD, those with CONUT scores of >3 were considered high risk [17,18]. This study was carried out in accordance with the principles of the Declaration of Helsinki. Our institution’s ethics committee approved the study’s protocol. Before study entry, the patients provided written informed consent that allowed the use of the data from their medical records only. The study was registered with the University Hospital Medical Information Network Clinical Trials Registry (UMIN000029848).

### 2.2. Treatment Protocols and Definitions

EVT was scheduled for patients who were diagnosed with PAD and who had a proven angiographic stenosis or an occlusion in a peripheral artery that was verified using imaging modalities, for example, enhanced computed tomography, magnetic resonance imaging or angiography. The EVT procedures were, in principle, performed using 6-Fr (guiding) sheaths for target lesions within the aortoiliac to popliteal arteries and 4.5-Fr guiding sheaths for isolated below-the-knee (BTK) lesions. Provisional stenting was performed on aortoiliac to popliteal lesions when flow limitations arose after balloon angioplasty that were caused by major arterial dissections, and balloon angioplasty alone was performed on BTK lesions in accordance with the healthcare reimbursement system in Japan. EVT that achieved at least “one straight line” to the vascular beds below the ankle was considered successful in the patients with critical limb ischemia (CLI). To prevent contrast-induced nephropathy, all patients with chronic kidney disease (CKD), except those on maintenance hemodialysis (HD), were given saline at a rate of 1.0 mL/kg/h for at least 12 h before the contrast was administered. EVT with carbon dioxide injections was considered on non-HD patients with end-stage renal dysfunction.

All patients received aspirin (100 mg) and/or clopidogrel (75 mg) daily for at least 1 month post-EVT. In principle, patients with newly implanted stents received both aspirin and clopidogrel. The durations of the administration of the antiplatelet and other medical treatment regimens were at the discretion of the attending physicians. Dermatologists and plastic surgeons independently undertook wound care and made decisions about amputations during their regular clinic visits. CLI was defined as chronic ischemic rest pain, ulcers, or gangrene that was attributable to objectively proven arterial occlusive disease [9]. Diabetes was defined as a fasting plasma blood glucose level of ≥126 mg/dL on 2 separate occasions or ≥200 mg/dL on any single occasion.

### 2.3. Data Collection and Assessment

We evaluated the patients’ baseline characteristics, including their ages, sexes, body mass indexes (BMIs), complete blood counts, which comprised the total white blood cell, neutrophil, and lymphocyte counts, and the hemoglobin levels, blood chemistry test results, which included the albumin, blood urea nitrogen (BUN), creatinine, sodium, C-reactive protein (CRP), brain natriuretic peptide (BNP), lipids, and hemoglobin A1c levels, and the estimated glomerular filtration rates (eGFRs), and the oral medications administered at the time of EVT. Data describing the patients’ Rutherford classifications and histories of smoking, hypertension, diabetes, dyslipidemia, CKD, and regular HD were collected from their medical records, and the target EVT lesions were documented based on the angiography findings collected from the medical records. The participants’ left ventricular ejection fractions (LVEFs) were determined using contrast ventriculography or echocardiography. The follow-up data were obtained from inpatient and outpatient medical records.

### 2.4. Study Endpoint

We compared the groups’ clinical and angiographic profiles, medications at the time of EVT, and long-term prognoses. The study’s primary endpoint was any death. The patients who had and did not have CLI and those who did and did not undergo regular HD were compared in relation to the study’s primary endpoint.

### 2.5. Statistical Analyses

Data are expressed as the means and the standard deviations or as numbers and percentages. After testing the distribution of each parameters by the Shapiro–Wilk test, none of them were normally distributed. Therefore, the non-parametric equivalent Mann–Whitney-U test were used to compare the two groups, and the analysis of variance and Kruskal–Wallis test were used to compare among the ≥3 groups with respect to the continuous variables, and the Chi-squared test and Fischer’s exact test were used to compare the groups with respect to the categorical variables, as appropriate. Kaplan–Meier analyses were performed to evaluate the differences among the groups in relation to all-cause mortality. Univariate and multivariate Cox regression analyses were performed to evaluate the associations between the baseline parameters, including the CONUT score categories, and the prognosis. For the multivariate analyses, age, sex, the laboratory data, including hemoglobin, eGFR, BUN, CRP, and BNP, LVEF, and CLI were used to adjust the models. Further, the additive discriminative ability of the CONUT score was evaluated by comparing the areas under the receiver operating characteristic (AUROCs) curves using binary techniques that involved 2 models that predict the primary endpoint, based on the method described by DeLong et al. [19]. To compare the AUROCs between the CONUT score and each of its components, the Z-test was applied. Two-tailed *p*-values < 0.05 were considered statistically significant. Statistical analyses were performed using R software version 3.3.0 (R Foundation for Statistical Computing, Vienna, Austria).

## 3. Results

### 3.1. Patients’ Baseline Characteristics

After excluding 77 patients whose laboratory data were incomplete, we evaluated data from 705 consecutive patients with PAD and for whom the CONUT scores could be calculated. Ultimately, 628 patients were enrolled to participate in this study (Figure 1). The patients were assigned to four risk groups based on their CONUT scores, as follows: low-risk group: *n* = 81 (12.9%); mild-risk group: *n* = 250 (39.8%); moderate-risk group: *n* = 169 (26.9%); and high-risk group: *n* = 128 (20.4%).

The patients’ mean CONUT score was 2.8 ± 2.2, and >67% of the patients had nutritional disturbances with CONUT scores of ≥2. Table 1 presents the patients’ baseline clinical characteristics. The patients’ mean age was 69 years. Male patients accounted for 69% and patients while CLI comprised 42% of the study population. Half of the study population underwent regular HD, and almost 70% of the patients had diabetes. In addition to differences with respect to the components of the CONUT score, significant differences were evident among the groups in relation to a range of parameters. The groups with higher CONUT scores contained more patients who were male, with lower BMIs, with CKD, who underwent HD, with impaired LVEFs, with histories of revascularization, and more patients with lower hemoglobin and low-density lipoprotein (LDL)-cholesterol levels and higher BUN, CRP, and BNP levels, and whose renal function was worse. The groups with higher CONUT scores had higher CLI rates. The administration of *β*-blockers and cilostazol at the time of EVT differed significantly among groups. Appendix A exhibits baseline characteristics in patients with and without CLI. Patients with CLI also had poorer clinical backgrounds compared to those without CLI. In patients classified with CONUT score, different trends were observed in sex, dyslipidemia, prior revascularization, WBC, neutrophil numbers, and EVT in below-the-knee lesions. Appendix A presents baseline clinical characteristics in patients with and without regular HD. Patients with HD generally had poorer clinical backgrounds compared to those without HD. When patients in each group were classified with CONUT score, different trends were observed in sex, BMI, prevalence of hypertension, impaired LVEF, neutrophil numbers, and prescription of β-blockers between patients with and without HD. 

### 3.2. Impact of the Conut Score on the Clinical Outcomes. 

During the 828-day follow-up period, 95 patients (15%) died. Figure 2 shows the Kaplan–Meier curves for all-cause mortality, with the curves representing the CONUT risk groups, that is, the nutritional status. The PAD patients in the group with the highest CONUT score had the lowest survival rates. The 2-year post-EVT survival rates in the groups with low, mild, moderate, and high CONUT scores were 97.3%, 92.2%, 86.5%, and 68.0%, respectively (*p* < 0.001; log-rank trend test). The Kaplan–Meier survival analyses showed that in the presence or absence of CLI, the patients with higher CONUT scores had lower survival rates compared with those in the patients with lower CONUT scores (*p* < 0.001; log-rank trend test) (Figure 3). Compared with the patients in the low-, mild-, and moderate-risk groups who did not have CLI, the patients in the high-risk group who did not have CLI had a markedly lower survival rate. Appendix A shows the causes of death. During the observation period, the death rate was more than two-fold higher for the patients with CLI (22.8%), compared with that for the patients who did not have CLI (9.6%). Systemic bacterial infection was the leading causes of death in both groups. Systemic bacterial infections, pneumoniae and sudden death in patients with CLI were significantly higher than those in patients with non-CLI (Appendix A). Additionally, the CLI patients with higher CONUT score (CONUT >5 and 3–4) also had higher amputation rate than those with lower CONUT score (CONUT 0–2) (log-rank trend test: *p* < 0.001) (Appendix A).

Appendix A shows Kaplan–Meier curves for all-cause mortality in patients with or without HD. The highest CONUT group (CONUT ≥ 5) had a poor prognosis compared with lower CONUT groups (CONUT 0–4) in PAD patients without HD. However, the higher CONUT score PAD patients with HD had the poorer prognosis.

### 3.3. Contribution of Each Conut Score Component to the Clinical Outcomes

An evaluation of the contribution of each component of the CONUT score to the primary endpoint showed that patients with higher scores for each component had lower survival rates (all *p* < 0.001; log-rank test) (Appendix A). Additionally, comparisons of the AUROC curves showed that the predictive value of the CONUT score (0.67; *p* < 0.001) was superior to that of any of the individual components of the CONUT score, namely, the lymphocyte count (0.64; *p* < 0.001), albumin level (0.59; *p* = 0.004), and the TC level (0.60; *p* < 0.001) (Appendix A).

### 3.4. Predictors of All-Cause Mortality. 

The univariate analysis determined values of *p* < 0.10 for age, sex, BMI, CLI, an LVEF <40%, hemoglobin, BUN, BNP, and eGFR (Table 2). The multivariate Cox hazard analysis determined that the CONUT score remained an independent predictor of all-cause mortality even after adjusting the model for multiple covariates (hazard ratio, 1.14; 95% confidence interval, 1.02–1.30) (Table 2).

## 4. Discussion

The principal finding from this retrospective analysis of patients with PAD who were undergoing EVT was that patients with a worse nutritional status were at a higher risk of all-cause mortality. Malnutrition is related to the severity of the atherosclerosis in patients with CAD and it predicts their mortality. The differences among risk groups regarding death became apparent soon after EVT, and they continued to increase during the observation period. This finding was independent of the presence of CLI. The CONUT score robustly represents a patient’s nutritional status, and it is a prognostic predictor in hospitalized patients. 

The CONUT score was also useful for predicting the long-term prognoses of patients who did and did not undergo regular HD. Evaluating the associations between nutritional status and serious clinical events in such a substantial number of high-risk patients, of whom 46% underwent regular HD and 71% had diabetes, may have helped to clarify the prognostic problems associated with EVT.

Prior studies investigated a variety of nutritional supplementation and indicators (BMI, Nutritional Risk Screening—NRS-2002, Malnutrition Universal Screening Tool—MUST, PNI, geriatric nutritional risk index—GNRI, CONUT score, among others) [20,21]. Guidelines recommend some of the above as standard nutritional indicators; however, these indices require many results from uncommon laboratory tests first, and little has been evaluated regarding their prognosis-predicting efficacy in atherosclerotic patients. Moreover, in terms of cost-effectiveness, the CONUT score only needs parameters to be collected routinely, in spite of its comparable predicting efficacy for long-term prognosis to full nutritional assessment [10].

Since PAD is a polyvascular disease and at an advanced stage of atherosclerosis, the level of systemic inflammation on PAD is higher than that on CAD [22]. Additionally, developing PAD has multidisciplinary mechanism involving inflammatory cytokines and oxidative stresses [23]. The presence of atherosclerosis causes malnutrition and the presence of malnutrition may be one of the risk factors developing atherosclerosis [20]. Therefore, the prognostic prediction of PAD patients should be assumed to evaluate both of nutritional status and the degree of inflammation. The CONUT score is not the nutritional guideline recommended tool [21]; however, this score might reflect both of the nutritional status and the degree of inflammation in atherosclerotic patients including PAD. As a result, this simple index successfully stratified short and long-term mortality risks of the advanced-staged atherosclerotic patients.

The AUROC curve analysis has shown that the predictive value of the CONUT score was better than that of its individual components. Lymphocyte risk score and TC risk score classification made differences in mortality rates among groups gradually, as time advanced after EVT. In contrast, the patients with high albumin risk scores showed marked declines in their mortality rates soon after EVT. Lymphocytes represent the systemic inflammatory status [24,25,26]. Many previous studies’ findings have shown that the lymphocyte count and the neutrophil/lymphocyte ratio can predict long-term mortality or amputations in CLI patients [26,27,28,29,30,31]. Our study’s findings concur with the findings from these studies, and they confirm the prognostic value of the lymphocyte count. The TC level, which was a complex of lipid profiles, might reflect the effect of not only chronic inflammation, but also a patient’s daily nutritional status, and it is related to digestive functions; therefore, it may have caused linear changes in the primary endpoint. Conversely, the serum albumin level may reflect multiple organ failure caused by excessive levels of free water and reductions in protein metabolism [32,33]. Therefore, patients in the high-risk albumin category may be at a higher risk of reduced survival rate during the early phase after EVT. A combination of parameters that predict short-term and long-term changes in the adverse event rate may be useful for risk stratifying PAD patients who have severe atherosclerotic profiles and whose prognoses are difficult to determine. 

Lowering of LDL-cholesterol levels appears to be associated with more favorable clinical prognoses for patients with atherosclerosis, including those with PAD and CLI [34,35,36]. However, findings from recent studies have shown worse outcomes in patients with malnutrition whose lipid profiles were controlled, which included low TC and triglyceride levels [37,38,39]. Since the triglyceride levels might reflect dietary habits, patients with advanced atherosclerosis and malnutrition, and especially those who undergo HD, should be treated carefully with lipid-lowering drugs.

Like previously reported findings [40,41], the leading cause of death in our study was systemic bacterial infection, the rate of which reached 30%, and it occurred at 30% in the patients with CLI who died and 25% in those without CLI. Bacterial infection in patients with severe atherosclerosis might be associated with any death. 

These results indicate that the impaired immune defenses caused by malnutrition may determine the long-term prognosis, irrespective of the severity of the atherosclerosis. If the nutritional status of the patients with PAD had been well controlled before the de novo development of CLI, their prognoses might have been more favorable. Therefore, the nutritional status of patients with PAD should be evaluated during the early stages of atherosclerosis, and risk stratifications for critical events should be performed properly.

The comorbidity associated with regular HD is one of the most prominent risk factors related to adverse clinical events [42,43]. Indeed, the large proportion of the patients (64.7%) who died in this study had undergone HD. HD patients are malnourished [44], which augments severe limb infections and leads to death or amputations. Predicting the prognoses of patients who undergo regular HD is difficult, because they have high rates of a diverse range of comorbidities. However, the CONUT score successfully stratified the risk of death in this study. In this very high-risk population, the prognostic value of the CONUT score was evident, even when these patients were compared with the patients who did not undergo HD. To date, an effective intervention has not been described that improves the clinical prognoses of patients with high CONUT scores. Hence, after determining the risks of adverse clinical events, nutritional supplementation and close observation should be considered for high-risk patients. To confirm this study’s findings, further interventional studies are warranted that will lead to improved outcomes after EVT.

The study population consisted of large numbers of patients who received regular HD, mainly because our institute has a huge dialysis center. The higher CONUT score group had higher prevalence of HD patients, and HD patients had higher mortality in this study. Given that the current study included the population at an extremely higher risk compared to that of prior trials and observations, it included a possible patient-selection bias, and it is not possible to generalize the findings as they are. However, among HD patients, the CONUT score at the time of EVT could stratify the risk of death during the long-term observation period, and even in the non-HD patients, it successfully extracted the high-risk population.

Collectively, the risk stratification of PAD patients, which was the population at an advanced stage of atherosclerosis, is meaningful for a better long-term prognosis, and the CONUT scoring system may be a promising option, independent of clinical presentation or HD, even it does not perfectly represent the ‘nutritional’ status in PAD patients. 

## 5. Study Limitations

This study was a single-center cohort with a relatively small sample size and a limited observation period as well as with a potential patient selection bias, and it included a limited population who had received EVT. The CONUT score was evaluated only at the time of EVT, and we did not assess the effect of the change of score during the observation period on the prognosis. Furthermore, it is unclear, based on this study’s results, whether the stratified risk classes provide an intervention.

## 6. Conclusions

Determining the nutritional status of patients by simply calculating the lymphocyte counts and the albumin and TC levels in patients undergoing EVT was associated with adverse clinical events, including all-cause mortality.

## Figures and Tables

**Figure 1 nutrients-11-01745-f001:**
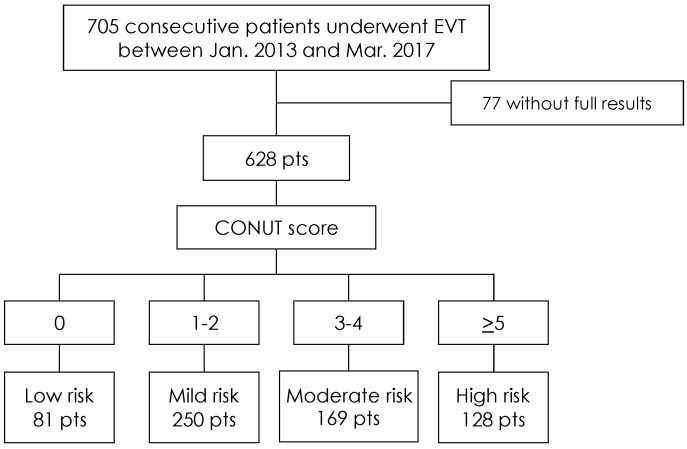
Study population. CONUT = Controlling Nutritional Status; EVT = endovascular therapy; pts = patients.

**Figure 2 nutrients-11-01745-f002:**
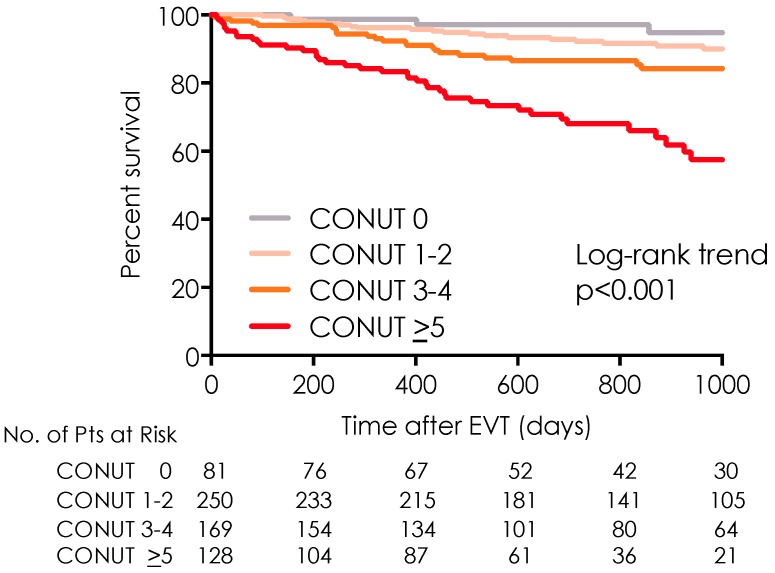
All-cause mortality after endovascular therapy. Kaplan–Meier curves of all-cause mortality among the four subgroups categorized according to the Controlling Nutritional Status score on admission. CONUT = Controlling Nutritional Status score; EVT, endovascular therapy.

**Figure 3 nutrients-11-01745-f003:**
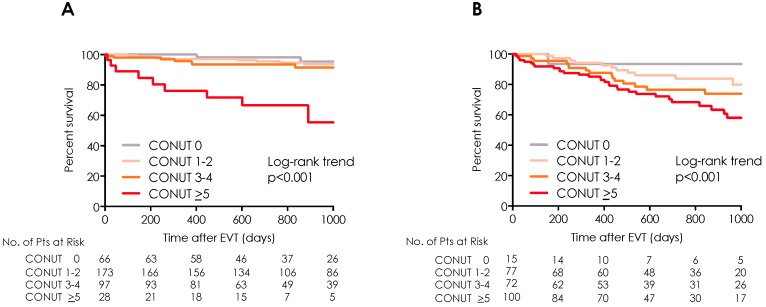
All-cause mortality after endovascular therapy in patients with and without chronic limb ischemia (CLI). Kaplan–Meier curves of the all-cause mortality rates for the four subgroups categorized according to the Controlling Nutritional Status score on admission in (**A**) Non-CLI patients and **(B)** CLI patients. CLI = critical limb ischemia; CONUT = Controlling Nutritional Status score; EVT = endovascular therapy.

**Table 1 nutrients-11-01745-t001:** Patients’ baseline characteristics.

Variables	Overall	Low-Risk Group	Mild-Risk Group	Moderate-Risk Group	High-Risk Group	*p*-Value
		(CONUT Score 0)	(CONUT Score 1–2)	(CONUT Score 3–4)	(CONUT Score ≥ 5)	
	*n* = 628	*n* = 81	*n* = 250	*n* = 169	*n* = 128	
Age, years	69 ± 10	70 ± 10	70 ± 11	68 ± 9	70 ± 10	0.11
>75 years old	205 (30)	27 (33)	87 (35)	45 (25)	46 (36)	0.26
Male	434 (69)	42 (52)	168 (67)	124 (73)	100 (78)	<0.001
BMI, kg/m^2^	23.3 ± 3.5	23.4 ± 3.2	23.9 ± 3.5	22.7 ± 3.4	22.9 ± 3.8	0.004
ABI	0.7 ± 0.6	0.7 ± 0.2	0.7 ± 0.2	0.7 ± 0.2	0.7 ± 0.3	0.87
Hypertension	520 (83)	62 (77)	219 (88)	135 (80)	104 (81)	0.06
Diabetes	445 (71)	54 (67)	169 (68)	125 (74)	97 (76)	0.29
Dyslipidemia	468 (75)	64 (79)	199 (80)	115 (68)	90 (70)	0.03
Smoking history	237 (38)	33 (41)	102 (41)	55 (33)	47 (37)	0.33
Prior PCI	305 (49)	29 (36)	118 (47)	82 (49)	76 (59)	0.01
Prior CABG	109 (17)	5 (6)	47 (19)	28 (17)	29 (23)	0.06
CKD †	474 (75)	43 (53)	178 (71)	138 (82)	115 (90)	<0.001
Hemodialysis	288 (46)	8 (10)	90 (36)	98 (58)	92 (72)	<0.001
LVEF<40%	73 (12)	3 (4)	18 (7)	23 (14)	29 (23)	<0.001
CLI	263 (42)	15 (19)	77 (31)	71 (42)	100 (78)	<0.001
Rutherford classification (1-3/4/5/6), n	359/40/212/17	66/2/11/2	169/15/63/3	96/16/55/2	28/7/83/10	<0.001
Lab data						
WBC, /μL	6917 ± 2553	7199 ± 1929	6884 ± 1960	6291 ± 2023	7629 ± 3975	<0.001
Neutrophil, /μL	4826 ± 2371	4492 ± 1738	4579 ± 1629	4527 ± 1822	5916 ± 3839	<0.001
Lymphocyte, /μL	1383 ± 569	2036 ± 371	1596 ± 499	1079 ± 362	955 ± 424	<0.001
Hemoglobin, g/dL	12.1 ± 1.8	13.4 ± 1.4	12.6 ±1.6	11.7 ± 1.7	11.0 ± 1.7	<0.001
Albumin, mg/dL	3.8 ± 0.6	4.1 ± 0.4	4.1 ± 0.4	3.8 ± 0.4	3.1 ± 0.5	<0.001
BUN, mg/dL	30 ± 17	22 ± 13	28 ±17	32 ± 15	38 ± 19	<0.001
Creatinine, mg/dL	4.1 ± 3.6	1.6 ± 2.3	3.6 ± 3.5	4.9 ± 3.6	5.6 ± 3.4	<0.001
eGFR, mL/min/1.73 m^2^	34 ± 30	57 ± 23	39 ± 30	28 ± 29	20 ± 24	<0.001
CRP, mg/dL	2.5 ± 10	3.3 ± 17	1.2 ± 6.7	1.6 ± 8.8	5.5 ± 11	<0.001
BNP, pg/dL	389 ± 641	112 ± 153	238 ± 434	438 ± 686	779 ± 868	<0.001
Total cholesterol, mg/dL	170 ± 40	210 ± 24	177 ± 34	157 ± 34	146 ± 43	<0.001
LDL-cholesterol, mg/dL	95 ± 31	124 ± 33	100 ± 27	86 ± 28	76 ± 26	<0.001
Triglyceride, mg/dL	137 ± 94	174 ± 90	140 ± 71	122 ± 68	125 ± 146	<0.001
HbA1c, %	6.7 ± 1.2	7.0 ± 1.2	6.7 ± 1.3	6.7 ± 1.4	6.5 ± 1.1	0.21
Target vessel						
Aorto-iliac	111 (17)	17 (21)	50 (20)	29 (17)	15 (12)	0.19
Femoro-popliteal	349 (56)	49 (60)	155 (62)	96 (57)	49 (38)	<0.001
Below the knee	275 (44)	21 (26)	87 (35)	70 (41)	97 (76)	<0.001
Medications						
Aspirin	399 (64)	45 (56)	167 (67)	102 (60)	85 (66)	0.21
Thienopyridine	356 (57)	40 (49)	142 (57)	103 (61)	71 (55)	0.37
Cilostazol	153 (24)	14 (17)	73 (29)	42 (25)	24 (19)	0.08
OAC	102 (16)	16 (20)	31 (12)	31 (18)	24 (19)	0.22
ACEi or ARBs	356(57)	48 (59)	152 (61)	90 (53)	66 (52)	0.21
βBlockers	273 (43)	19 (23)	104 (42)	76 (45)	74 (58)	<0.001
Statins	330 (53)	48 (59)	130 (52)	87 (51)	65 (51)	0.64

The data presented are the numbers (%) or the means and the standard deviations. ACEi = angiotensin-converting enzyme inhibitor; ARB = angiotensin II receptor blocker; BMI = body mass index; BNP = brain natriuretic peptide; BUN = blood urea nitrogen; CABG = coronary artery bypass grafting; CLI = critical limb ischemia; CKD = chronic kidney disease; CONUT = Controlling Nutritional Status; CRP = C-reactive protein; DM = diabetes mellitus; eGFR = estimated glomerular filtration rate; HbA1c = hemoglobin A1c; LDL = low-density lipoprotein; LVEF = left ventricular ejection fraction; OAC = oral anticoagulant; PCI = percutaneous coronary intervention; WBC = white blood cell. † eGFR < 60 mL/min.1.73 m^2^.

**Table 2 nutrients-11-01745-t002:** Univariate and multivariate analyses of any death.

Variables	Univariate			Multivariate		
	HR	95% CI	*p*-Value	HR	95% CI	*p*-Value
CONUT score	1.31	1.21–1.41	<0.001	1.14	1.02–1.30	0.03
Age	1.03	1.01–1.05	0.01	1.04	1.01–1.07	0.003
Male	1.61	0.99–2.60	0.05	1.34	0.79–2.29	0.28
BMI	0.93	0.88–0.99	0.02	0.99	0.93–1.06	0.93
CLI	3.06	2.01–4.65	<0.001	1.32	0.79–2.21	0.29
Diabetes	1.08	0.70–1.69	0.72			
LVEF < 40%	3.23	2.09–5.17	<0.001	1.98	1.20–3.29	0.008
Hemoglobin	0.74	0.67–0.83	<0.001	0.95	0.81- 1.10	0.49
eGFR	0.98	0.97–0.99	<0.001	0.99	0.98–1.00	0.07
CRP	1.01	0.99–1.03	0.24			
BNP. log	4.51	3.09–6.58	<0.001	2.78	1.72–4.44	<0.001

BMI = body mass index; BNP = brain natriuretic peptide; BUN = blood urea nitrogen; CI = confidence interval; CLI = critical limb ischemia; CONUT = Controlling Nutritional Status; CRP = C-reactive protein; eGFR = estimated glomerular filtration rate; HR = hazard ratio; LVEF = left ventricular ejection fraction.

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
