# Peer review of "The Baseline Nutritional Status Predicts Long-Term Mortality in Patients Undergoing Endovascular Therapy"

_nutrients, 2019, doi:10.3390/nu11081745_

Round 1

Reviewer 1 Report

The paper is very interesting.

Although this reviewer warmly welcomes this interesting manuscript, some concerns should be addressed:

The rationale for the study is unclear as the introduction is a bit confusing, comprising several pieces of apparently unlinked information. A more integrated appraisal of the relevant literature would be appropriate to provide the context for the study.

The updated epidemiology of PAD should be mentioned in the introduction (Shu et al. Atherosclerosis. 275:379-381).

Statistical analysis: The Authors did not determine whether sample values are normally distributed (e.g. Kolmogorov-Smirnov Test) before applying the Student’s t test.

The strengths and limitations of the study should be deeply addressed, taking into account sources of potential bias or imprecision: Discuss both direction and magnitude of any potential bias. 

Author Response

Response to Reviewers

Point-by-point responses to the reviewers’ original comments shown in bold and red have been addressed as follows. All the changes in the manuscript are in based on comments from Editor, Reviewer #1, and Reviewer #2.

We deeply appreciate editors’ giving a chance for revision. We have carefully revised the manuscriptin consideration of the suggestions of the reviewers. We are looking forward to hearing your favorable evaluation.

Reviewer #1: 

Question1: The rationale for the study is unclear as the introduction is a bit confusing, comprising several pieces of apparently unlinked information. A more integrated appraisal of the relevant literature would be appropriate to provide the context for the study. The updated epidemiology of PAD should be mentioned in the introduction (Shu et al. Atherosclerosis. 275:379-381).

ResponseRegarding with reviewer’s comment, wehave newly added references and made modifications in the Introduction part (Page 5, Line 1).

Question2: The Authors did not determine whether sample values are normally distributed (e.g. Kolmogorov-Smirnov Test) before applying the Student’s t test.

Response: We used Kolmogorov-Smimov test to evaluate the distribution of the samples. As results, all parameters evaluated in this study were not normally distributed. Therefore, we have newly added this information and made corrections in the Statistical analysis part (Page 10, Line 9).

Question3: The strengths and limitations of the study should be deeply addressed, taking into account sources of potential bias or imprecision: Discuss both direction and magnitude of any potential bias. 

Response: The study population consisted of large numbers of patients who received regular HD, mainly due that our institute had a huge dialysis center. Higher CONUT score group had higher prevalence of HD patients, and HD patients had higher mortality in this study. Given the current study included the population at an extremely higher risk compared to that in prior trials and observation, it included a possible patient-selection bias, and it is not possible to generalize the findings as they are. However, among HD patients, CONUT score at the time of EVT could stratify the risk of death during the long-term observation period, and even in the non-HD patients, it successfully extracted the high-risk population. Collectively, the risk stratification of PAD patients, that was the population at an advanced stage of atherosclerosis, is meaningful for the better long-term prognosis, and CONUT scoring system may be a promising option, independent of clinical presentation or HD, even it does not perfectly represent the ‘nutritional’ status in PAD patients. As reviewer’s suggestions, we have newly added the comments above focusing on the bias in this study in the Discussion part (Page 22, Line 1).

Reviewer 2 Report

Summary

This is an interesting as well as relevant topic discussing the prognostic impact of a nutrition-risk score on patients with peripheral artery disease. 

To my knowledge, neither the CONUT score nor any of its components are recommended for malnutrition assessment in current international guidelines as each component is confounded by comorbidities and do not represent markers of nutrition. Even though the chosen score was prognostically relevant in this cohort study, it would seem more useful to compare this score to the current standards regarding the screening and assessment of malnutrition.

The quality of the English is very good, only minor spellchecks and minor re-formatting are required. The tables and figures are relevant to the content and clear in design.

Broad comments 

This groups reason to use a nutrition-risk score was derived from patients with vascular disease, such as coronary artery disease and heart failure. However, the pathomechanisms share similarities, as well as differences and in my opinion, the manuscript would benefit from a little more detailed explanation on the connection between and relevance of (mal)nutrition and peripheral artery disease.

Why was only the CONUT score evaluated here? None of the components of the CONUT score are recommended by international guidelines regarding the evaluation of malnutrition? The CONUT score seems useful to predict the prognosis, but as the authors discuss themselves, the score might not have uch to do with nutrition, but comorbidities and overall severity of disease.

A comparison between different nutrition-risk scores might have given more insights.

I also suggest giving a little more background and detail on patients with regular hemodialysis, as they seem to be very frequent in this patient group.

Specific comments

·       Pg 1, line 23&25: in what period of time?

·       Pg 1, line 24-25: consider adding p-values

·       Pg 1, line 34: consider explaining which comorbidities are relevant for the prognosis

·       Pg 2, lines 42-44: redundand with methods section

·       Pg 4, line 129: Half of the patients undergoing HD seems to be very frequent – can you give an explanation for this great proportion, or introduce a bit more in the background?

·       Pg 7, line 10: was the study retrospective, if written informed consent was obtained prior to study entry?

·       Pg 7, lines 209-225: see my comment above regarding the selection of the score and its individual components. Were any other validated scores determined and used as well?

Author Response

Response to Reviewers

Point-by-point responses to the reviewers’ original comments shown in bold and red have been addressed as follows. All the changes in the manuscript are in based on comments from Editor, Reviewer #1, and Reviewer #2.

We deeply appreciate editors’ giving a chance for revision. We have carefully revised the manuscriptin consideration of the suggestions of the reviewers. We are looking forward to hearing your favorable evaluation.

Reviewer #2: 

This groups reason to use a nutrition-risk score was derived from patients with vascular disease, such as coronary artery disease and heart failure. However, the pathomechanisms share similarities, as well as differences and in my opinion, the manuscript would benefit from a little more detailed explanation on the connection between and relevance of (mal)nutrition and peripheral artery disease.

Response: Because PAD is a polyvascular disease and at an advanced stage of atherosclerosis, the level of systemic inflammation on PAD is usually higher than that on CAD. Additionally, developing PAD has multidisciplinary mechanism involving inflammatory cytokines and oxidative stresses. The presence of atherosclerosis causes malnutrition and the presence of malnutrition may be one of the risk factors developing atherosclerosis. Therefore, the prognostic prediction of PAD patients should be assumed to evaluate both of nutritional status and the degree of inflammation. CONUT score is not the nutritional guideline recommended tool; however, this score might reflect both of the nutritional status and the degree of inflammation in atherosclerotic patients including PAD. As results, this simple index successfully stratified short and long-term mortality risks of the advanced-staged atherosclerotic patients. As reviewer’s suggestions, we have newly added the sentence above in the manuscript, and placed appropriate references (Page 18, Line 1, reference No. 20, 22, 23).

Why was only the CONUT score evaluated here? None of the components of the CONUT score are recommended by international guidelines regarding the evaluation of malnutrition? The CONUT score seems useful to predict the prognosis, but as the authors discuss themselves, the score might not have uch to do with nutrition, but comorbidities and overall severity of disease.

Response:We agree with reviewer’s opinions. Recently, we have diverse parameters or indices evaluating patients’ nutritional status. Prior studies investigated a variety of nutritional supplementation and indicators (BMI, NRS-2002, MUST, PNI, GNRI, CONUT score, and more). Guidelines recommend some of above as standard nutritional indicators; however, these indices more or less need many results from uncommon laboratory tests, and little has been evaluated their prognosis-predicting efficacy in atherosclerotic patients. Moreover, in terms of cost-effectiveness, CONUT score only needs routinely collecting parameters, in spite of its comparable predicting efficacy for long-term prognosis to full nutritional assessment. As reviewer’s suggestions, we have newly added the sentence above in the manuscript, and placed appropriate references (Page 17, Line 7, reference No. 20, 21).

I also suggest giving a little more background and detail on patients with regular hemodialysis, as they seem to be very frequent in this patient group.

Response: We appreciated reviewer’s very interesting point of view. The study population consisted of large numbers of patients who received regular HD, mainly due that our institute had a huge dialysis center. Higher CONUT score group had higher prevalence of HD patients, and HD patients had higher mortality in this study. Given the current study included the population at an extremely higher risk compared to that in prior trials and observation, it included a possible patient-selection bias, and it is not possible to generalize the findings as they are. However, among HD patients, CONUT score at the time of EVT could stratify the risk of death during the long-term observation period, and even in the non-HD patients, it successfully extracted the high-risk population. Collectively, the risk stratification of PAD patients, that was the population at an advanced stage of atherosclerosis, is meaningful for the better long-term prognosis, and CONUT scoring system may be a promising option, independent of clinical presentation or HD, even it does not perfectly represent the ‘nutritional’ status in PAD patients. As reviewer’s suggestions, we have newly added baseline clinical profiles of HD and Non-HD patients in the Supplemental Table and the sentence above in the manuscript, and placed appropriate references (Page 22, Line 1).

Specific comments

·       Pg 1, line 23&25: in what period of time?

Response: In this study population, patients with higher CONUT score had worse clinical profiles on admission. For better understanding of readers, we have modified the sentence, as reviewer suggested (Page 3, Line 16).

·       Pg 1, line 24-25: consider adding p-values

Response: P values were added in each part, according to reviewer’s instructions (Page 3, Line 15-16).

·       Pg 1, line 34: consider explaining which comorbidities are relevant for the prognosis

Response: Regarding with Reviewer’s suggestion, we modified the sentence in the Introduction part (Page 5, Line 2).

·       Pg 2, lines 42-44: redundand with methods section

Response: As reviewer’s suggestion, we moved the explanation and definition of CONUT score to the Methods section.

·       Pg 4, line 129: Half of the patients undergoing HD seems to be very frequent – can you give an explanation for this great proportion, or introduce a bit more in the background?

Response: The study population consisted of large numbers of patients who received regular hemodialysis, mainly due that our institute had huge dialysis center. Higher CONUT score group had higher prevalence of HD patients, and HD patients had higher mortality in this study. It means that the current study included the population at higher risk than that of prior studies and trials. Given the current study included the population at an extremely high risk, it included a possible patient selection bias, and it is not possible to generalize the findings as they are. However, even in the non-HD patients, CONUT score at the time of EVT successfully stratified the risk of death during the long-term observation period. We have added this new result in the Supplemental materials, and rewritten the manuscript in the Results and the Discussion part (Page 13, Line 7, Page 22, Line 1)

·       Pg 7, line 10: was the study retrospective, if written informed consent was obtained prior to study entry?

Response: As described in the manuscript, we routinely obtained informed consent prior to the study entry in all patients who received EVT at least for recorded data use only for the observational cohort study.

·       Pg 7, lines 209-225: see my comment above regarding the selection of the score and its individual components. Were any other validated scores determined and used as well?

Response: As mentioned above, CONUT scoring system has simple calculation methods as well as sufficient prognostic prediction ability. Therefore, we selected CONUT score at the time of EVT for assessing patient systemic status influencing long-term mortality. We have added the explanation in the Discussion part (Page 17, Line 7).